# Seroprevalence and Molecular Characterization of *Coxiella burnetii* in Cattle in the Republic of Korea

**DOI:** 10.3390/pathogens9110890

**Published:** 2020-10-27

**Authors:** Sunwoo Hwang, Hyung-Chul Cho, Seung-Uk Shin, Ha-Young Kim, Yu-Jin Park, Dong-Hoon Jang, Eun-Mi Kim, Jong Wan Kim, Jinho Park, Kyoung-Seong Choi

**Affiliations:** 1Department of Horse/Companion and Wild animals, College of Ecology and Environmental Science, Kyungpook National University, Sangju 37224, Korea; hdh123111@naver.com (S.H.); ujp1506@naver.com (Y.-J.P.); janginsect@naver.com (D.-H.J.); 2Department of Animal Science and Biotechnology, College of Ecology and Environmental Science, Kyungpook National University, Sangju 37224, Korea; gudcjf246@naver.com (H.-C.C.); shinws95@naver.com (S.-U.S.); kimem256@naver.com (E.-M.K.); 3Bacterial Disease Division, Animal and Plant Quarantine Agency, Gimcheon 39660, Korea; kimhy@korea.kr (H.-Y.K.); biotics@korea.kr (J.W.K.); 4College of Veterinary Medicine, Jeonbuk National University, Iksan 54596, Korea; jpark@jbnu.ac.kr

**Keywords:** *Coxiella burnetii*, dairy cattle, beef cattle, grazing, ELISA, *IS1111*

## Abstract

This study was conducted to determine the prevalence of *Coxiella burnetii* in cattle and how that prevalence is influenced by cattle breed and growth type. A total of 491 cattle [cattle breed: 216 dairy cattle and 275 beef cattle; growth type: indoor housed (*n* = 294) and grazing (*n* = 197)] were used. The presence of *C. burnetii* DNA and antibodies was detected from blood and serum samples using polymerase chain reaction (PCR) and enzyme-linked immunosorbent assay (ELISA), respectively. The overall prevalence of *C. burnetii* was: 10.8% (95% CI: 8.0–13.5%) using PCR and 8.8% (95% CI: 6.3–11.3%) using ELISA. The prevalence of *C. burnetii* was significantly higher in beef cattle than in dairy cattle using both PCR (13.5% vs. 7.4%; *P* = 0.032) and ELISA (14.5% vs. 1.4%; *P* = 0.000), respectively. Comparison by growth type revealed that *C. burnetii* infection was significantly higher in grazing cattle than in housed cattle when using both PCR (24.9% vs. 1.4%; *P* = 0.000) and ELISA (21.3% vs. 0.3%; *P* = 0.000). Beef cattle were at a significantly higher risk of contracting *C. burnetii* compared with dairy cattle (odds ratio = 3.20, 95% CI: 1.80–5.67; *P* = 0.000). The risk of contracting *C. burnetii* in grazing cattle was increased by 32.57-fold (95% CI: 12.84–82.61; *P* = 0.000) compared with indoor housed cattle. The phylogenetic analysis based on the IS1111 gene revealed that our sequences grouped with human, tick, goat, and cattle isolates/strains found in several countries. *C. burnetii* sequences circulating in the Republic of Korea exhibit genetic variations. Thus, grazing is a high risk factor for the prevalence and transmission of *C. burnetii*.

## 1. Introduction

*Coxiella burnetii*, the causative agent of Q fever, is a highly infectious zoonotic intracellular bacterium that can infect a wide range of hosts including wild and domestic animals, birds, and arthropods [1,2,3]. Transmission to humans is primarily caused by the inhalation of contaminated aerosols or dust, or from direct contact with infected animals (mostly sheep and goats) [4]. Infected animals are often asymptomatic; however, *C. burnetii* infection is associated with abortion and stillbirths in sheep and goats, and infertility and endometritis in cattle [5,6]. The bacteria in infected animals can be shed in vaginal discharge, urine, feces, semen, milk, and birth products (placenta and birth fluids) [7,8]. More importantly, the shedding of *C. burnetii* in milk poses a potentially significant threat to public health since the consumption of raw milk and unpasteurized milk products is still practiced; thus, this could be a source of human infections [9,10]. Q fever has been underestimated in humans due to the difficulty in its diagnosis and its relatively asymptomatic manifestation. Nevertheless, Q fever is a public health concern as it ranks as one of the thirteen leading global priority zoonoses. Furthermore, it has been considered as a potential biological weapon due to its widespread availability, aerosolized use, and environmental stability [11,12]. 

The diagnosis of Q fever in livestock is very difficult because of its nonspecific clinical symptoms [2]. Exposure to *C. burnetii* and zoonotic risks in cattle have generally been determined using serological surveys in many countries [13,14,15,16]. Incidentally, seropositivity to *C. burnetii* is not strongly correlated with the shedding of the bacterium. Although serological analyses cannot be used to determine actual contamination in a herd, it is a valuable tool for the screening of *C. burnetii* infection within herds. Recent studies in the Republic of Korea (ROK) have shown that the overall seroprevalence was 10.5% in cattle and 19.1% in Korean native goats (*Capra hircus coreanae*) [14,17], indicating that the prevalence of *C. burnetii* is of significance to domestic ruminants in the regions. Korean cattle breeding herds rank 65th in the world and their population size has been gradually increasing. Moreover, meat consumption is steadily increasing as a result of westernized and improved nutrition and living conditions. Annual meat consumption per capita is an average of 47.6 kg in the ROK with a tendency of raw meat consumption in some parts of the population. This practice is a risk factor associated with the transmission of pathogens found in meat. Despite its zoonotic potential, little is known about the importance and risk factors associated with *C. burnetii* transmission and infection in cattle in the ROK. Therefore, the objective of this study was to evaluate the prevalence of *C. burnetii* and its association with cattle breed and growth type as well as characterization of the genetic diversity of *C. burnetii* circulating in the ROK.

## 2. Results

### 2.1. Detection of C. burnetii Using Molecular and Serologic Assays

The results of the polymerase chain reaction (PCR) and enzyme-linked immunosorbent assay (ELISA) of *C. burnetii* in the breeds within each growth type are presented in Table 1. The overall prevalence of *C. burnetii* in the cattle was 10.8% (95% confidence interval (CI): 8.0–13.5%) and 8.8% (95% CI: 6.3–11.3%) using the PCR analysis and ELISA serological test, respectively. As shown in Table 2, the prevalence of *C. burnetii* was significantly higher in beef cattle than in dairy cattle using PCR (13.5% vs. 7.4%; *P* = 0.032) and ELISA (14.5% vs. 1.4%; *P* = 0.000). With regard to the growth types, the prevalence of *C. burnetii* was significantly higher in grazing cattle (PCR: 24.9%, 95% CI: 18.8–30.9%; ELISA: 21.3%, 95% CI: 15.6–27.0%) than in housed cattle (PCR: 1.4%, 95% CI: 0–2.7%; ELISA: 0.3%, 95% CI: 0–1.0%).

### 2.2. Risk Factor Analysis

Of the growth types, grazing exhibited a significant effect on the prevalence of *C. burnetii*. In grazing cattle, *C. burnetii* was detected at 33.34-fold higher using PCR analysis (95% CI: 11.77–94.80; *P* = 0.000) and 114.51-fold higher using the ELISA test (95% CI: 15.51–842.41; *P* = 0.000), respectively, compared to housed cattle (Table 3). Based on PCR or ELISA positivity for *C. burnetii*, the association of the selected risk factors for *C. burnetii* and its prevalence is shown in Table 4. The results indicated that beef cattle (odds ratio (OR) = 3.20, 95% CI: 1.80–5.67; *P* = 0.000) were at a significantly higher risk of contracting *C. burnetii* infection than dairy cattle. When the cattle were permitted to graze, the risk of contracting *C. burnetii* infection increased 32.57-fold (95% CI: 12.84–82.61; *P* = 0.000) compared with housed cattle (Table 4).

### 2.3. Phylogenetic Analysis ofIS1111 Sequences

To investigate the genetic relationship among *C. burnetii* sequences detected in dairy and beef cattle, a total of 53 PCR-positive samples were sequenced. Since the sequences from these samples were similar, 13 sequences that were relatively different from the rest were selected and used for phylogenetic analysis. The *C. burnetii* sequences exhibited 95.6%–99.5% homology to one another. A phylogenetic tree constructed from the partial 202 bp gene sequences revealed that the *C. burnetii* sequences found in beef and dairy cattle were clustered with several strains of *C. burnetii* obtained from ticks, humans, goats, and cattle from other countries (Figure 1). Interestingly, cattle sequences obtained in this study shared a similarity of 93.7–97.1% with that of the Korean water deer (*Hydropotes inermis argyropus*) recently found by our group. 

## 3. Discussion

In the present study, the overall prevalence of *C. burnetii* in cattle was determined to be 10.8% using PCR and 8.8% using ELISA. The seroprevalence of this result was similar to that of another study performed in the ROK [14]. Seroprevalence for *C. burnetii* in cattle varied from that reported in other countries: 6.7% in Spain [18], 6.8% in Nigeria [19], 10.5% in Kenya [20], 11% in Algeria [15] and Iran [21], 14.4% in Italy [22], 16.3% in the East of Turkey [23], 19.3% in Egypt [24], 25.4% in Poland [13], 29.1% in India [11], 31.3% in Cameroon [25], and 42.9% in Ecuador [26]. This variation in seroprevalence among countries may be explained by the herd management systems. Furthermore, the prevalence of *C. burnetii* infection detected by PCR analysis reported in this study was higher than that reported in another study conducted in the ROK [14]. The difference between the two groups may be explained by the number of cattle sampled (491 in this study vs. 736 in Seo et al. [14]), regions of selected farms, and variations in the target gene used for detection. The *IS1111* PCR assay conducted in this study has been proven to be highly specific and sensitive for the detection of *C. burnetii* in various clinical samples [27,28]. Our PCR results were similar to those obtained in Iran (7.5%) and Zambia (7.7%) [21,29].

Our findings showed that the seroprevalence of *C. burnetii* was 1.4% in dairy cattle and 14.5% in beef cattle. To date, studies on *C. burnetii* have been primarily conducted in dairy cattle [13,30,31,32,33] because it is believed that milk consumption increases the risk of *C. burnetii* transmission, posing a significant threat to public health. In this study, an almost ten-fold higher seroprevalence of *C. burnetii* was observed in beef cattle (Table 2). Although we cannot make a precise determination at this point, the higher seroprevalence in beef cattle than that in dairy cattle is attributable to the difference in farm management systems, rather than the difference in cattle breed. 

*Coxiella *burnetii** DNA was detected using PCR to evaluate blood samples from both beef and dairy cattle. In dairy cattle, the presence of *C. burnetii* DNA was much higher than the results of antibodies against *C. burnetii*, whereas in beef cattle, the prevalence of *C. burnetii* was significantly high using both the PCR (*P* = 0.032) and ELISA (*P* = 0.000) methods. PCR analysis has the advantage of detecting an ongoing infection. Since all cattle examined in this study appeared healthy due to the lack of clinical signs, we did not expect that these animals would be infected with *C. burnetii*. Nevertheless, the PCR results indicate the possibility of the cattle shedding the pathogen through milk, urine, and feces, making them a potential source of human infection. In addition, some parts of the ROK population consume raw beef, representing a significant public health concern. Although the presence of bacteremia in cattle that tested positive for PCR has not been established, further studies are necessary to investigate its association with clinical symptoms through bacteria isolation. 

We found that the prevalence of *C. burnetii* was much higher in cattle on pasture than in housed cattle. This finding is consistent with that of Capuano et al., who reported that the prevalence of *C. burnetii* was higher in grazing cattle (19.6%) than that in housed cattle (13.2%) [22]. In contrast, the prevalence of *C. burnetii* in the present study was 36.0% and 1.7% in grazing cattle and housed cattle, respectively, which was much higher than that reported by Capuano et al. [22]. The reason why our results were high in grazing cattle may be explained by the method of detection. ELISA and PCR used in the present study are reliable diagnostic methods for detecting *C. burnetii* infection. We used both methods and considered a positive result if PCR or ELISA was positive; hence, we believe that there was a difference in the prevalence. There was not much difference compared with the results of seroprevalence. In addition, the low prevalence of *C. burnetii* in housed cattle seen in the current study may reflect differences in the management of the cattle. The risk was highest when cattle were housed after grazing; this was consistent with our results [22]. Grazing systems have many advantages including animal welfare [34], but there is a higher risk of contracting tick-borne diseases because of increased exposure to ticks [35]. Due to global warming, the climate of the ROK has become subtropical, and as a consequence, tick species are expanding their geographical range and are becoming a growing concern for human and animal health [36]. *Haemaphysalis longicornis* is the predominant tick species in the ROK. According to a recent report, *Coxiella*-like endosymbionts have been found in *H. longicornis* parasitizing horses [37]. Ticks shed significant loads of *C. burnetii* in their feces and saliva, and may be another potential source of bacterial transmission [38]. In light of this, the transmission of *C. burnetii* by ticks cannot be discounted. Tick infestation in the grazing cattle was observed in the current study, but the likelihood of the ticks being *C. burnetii* vectors was not investigated, thus the route of transmission in these cattle remains unknown. Additional epidemiological studies are required to investigate the prevalence of *C. burnetii* in tick species present in the ROK to determine their contribution to pathogen transmission. 

Generally, dairy cattle are less likely to graze than beef cattle in the ROK. Although the sample size in the current study was small, the prevalence of *C. burnetii* in grazing dairy cattle was relatively high. An additional mode of *C. burnetii* transmission is the shedding of the pathogen via the fecal matter and urine of infected animals on pasture. The bacteria can remain in the environment for extended periods and may be aerosolized, becoming a potential hazard to humans and animals [21]. Spores of *C. burnetii* can spread over several kilometers from the primary source of infection due to wind action, increasing the exposure of animals that are reared on pasture [39]. When cattle graze on a pasture, *C. burnetii* can be transmitted through the inhalation of contaminated aerosols or dust (the most important mode of transmission), rather than ticks, leading to infection. The possible interaction of grazing cattle with wildlife presents yet another risk for the transmission of *C. burnetii*. A recent study performed by our group reported that *C. burnetii* was identified in Korean water deer [40], making them potential reservoirs for this bacterium. Thus, we can conclude that grazing cattle are more susceptible to *C. burnetii* infection than housed cattle. 

According to our results, *C. burnetii* sequences identified in beef and dairy cattle using the *IS1111* gene exhibited a slightly different sequence homology from that of Korean water deer previously reported by our group. Our findings indicate that genetic variation exists in *C. burnetii* sequences circulating in the ROK. Interestingly, the cattle sequences shared 95.6–100% similarity with pathogenic *C. burnetii* strains isolated from *Hyalomma dromedarii* in Tunisia, humans in Greece, and cattle in France [41]. These findings suggest that *C. burnetii* sequences detected in the ROK are probably zoonotic and pathogenic. Additional molecular epidemiological studies are needed to further investigate the genetic diversity of this bacterium in humans and animals alike.

In conclusion, the present study demonstrates that beef cattle have a higher prevalence of *C. burnetii* and are at more risk than dairy cattle. In particular, grazing is a higher risk factor in the transmission of the infection to animals. The prevalence of *C. burnetii* is of public health concern and poses a significant risk to humans that come in close contact with animals. Our findings seek to increase awareness on the importance of *C. burnetii* as a potential zoonotic pathogen of grazing cattle in the ROK. These results provide useful information for a better understanding of the prevalence of *C. burnetii* and also for designing disease control strategies for cattle. Further studies should be aimed at evaluating potential transmission risks and the pathogenicity of *C. burnetii* circulating in the ROK.

## 4. Materials and Methods

### 4.1. Ethical Statement

All animal procedures were performed according to the ethical guidelines for the use of animal samples as permitted by Jeonbuk National University (Institutional Animal Care and Use Committee decision No. CBU 2014-00026). All procedures and possible consequences were explained to the farm owners/managers at the surveyed farms. Written informed consent was obtained for the collection of blood samples from the cattle owners.

### 4.2. Experimental Design

Farms were classified according to the growth type practiced as being either a grazing type or a housed type. Grazing cattle were allowed to graze from spring to fall and housed in the winter. Housed cattle were raised only indoors without pasturing; food and water were provided, and movement was restricted due to space limitations. The cattle were further divided into four subgroups: dairy/grazing, dairy/housed, beef/grazing, and beef/housed (Table 1).

### 4.3. Blood Sample Collection

Approximately 10 mL blood samples were collected from the jugular veins of 491 cattle (216 dairy cattle and 275 beef cattle) from five different regions of the ROK (Figure 2). Blood was equally divided into an anti-coagulation collection tube (BD Vacutainer^®^, Franklin Lakes, NJ, USA) and a SST tube (BD Vacutainer^®^), and then delivered to the laboratory. The serum was separated and collected by centrifugation at 3000× *g* for 20 min and then stored at −20 °C until use. Whole blood was used for DNA extraction and serum was used for serology. All animals were clinically healthy.

### 4.4. DNA Extraction and PCR

DNA was extracted from 200 µL of each blood sample using the DNeasy Blood Kit (Qiagen, Valencia, CA, USA) according to the manufacturer’s instructions and stored at −80 °C. Screening for the *C. burnetii* was performed using the *IS1111* transposase insertion element [42]. PCR conditions were 93 °C for 3 min, followed by 30 cycles of 93 °C for 30 s, annealing at 54 °C for 30 s, and 72 °C for 1 min. For each PCR run, negative and positive controls were included. The size of the amplified fragment was 202 bp. Secondary PCR products were separated by electrophoresis on 1.5% agarose gel and visualized after staining with ethidium bromide.

### 4.5. Serological Screening of Serum Samples

Serum samples from the cattle were tested for antibodies against *C. burnetii* using a commercial ELISA kit (ID Screen® Q fever Indirect Multi-species kit; ID.vet, Gabriels, France) according to the manufacturer’s instructions. For normalization of the optical density (OD) results, the sample/positive control (S/P) ratio was calculated for each sample as follows: Value (%) = (OD sample − OD negative control)/(OD positive control − OD negative control) × 100. Samples with an S/P% greater than 50% were considered positive; between 40% and 50%, doubtful; and less than 40%, negative. In this study, doubtful results were considered negative.

### 4.6. Phylogenetic Analysis

All secondary PCR products were purified using the AccuPower PCR Purification Kit (Bioneer, Daejeon, Korea) and used for direct sequencing (Macrogen, Daejeon, Korea). The nucleotide sequences obtained in this study were aligned using ClustalX and compared with the reference sequences from the GenBank database. A phylogenetic tree was constructed based on the *IS1111* fragments using the maximum-likelihood method in the MEGA7 software [43]. The reliabilities of the tree were assessed using a bootstrap analysis with 1000 replicates.

### 4.7. Statistical Analysis

Statistical analyses were performed using chi-square test in the SPSS Statistics version 25 software package for Windows (SPSS, Inc., Chicago, IL, USA). The 95% CI was estimated using the following formula:$$P\;\pm \;1.96\;\times \sqrt{\frac{P\left(100\;-\;P\right)}{N}}\;(\%),\;(p\;=\;\mathrm{prevalence},\;n\;=\;\mathrm{sample}\;\mathrm{size})$$

A univariable multinomial logistic regression analysis was performed to determine the detection rate of *C. burnetii* in dairy and beef cattle depending on the growth type. In addition, an analysis of the risk factors (cattle breed and growth type) associated with *C. burnetii* infection was performed using a univariable logistic regression model. The OR and 95% CI were determined. A *P*-value of ≤0.05 was considered to be statistically significant. 

## Figures and Tables

**Figure 1 pathogens-09-00890-f001:**
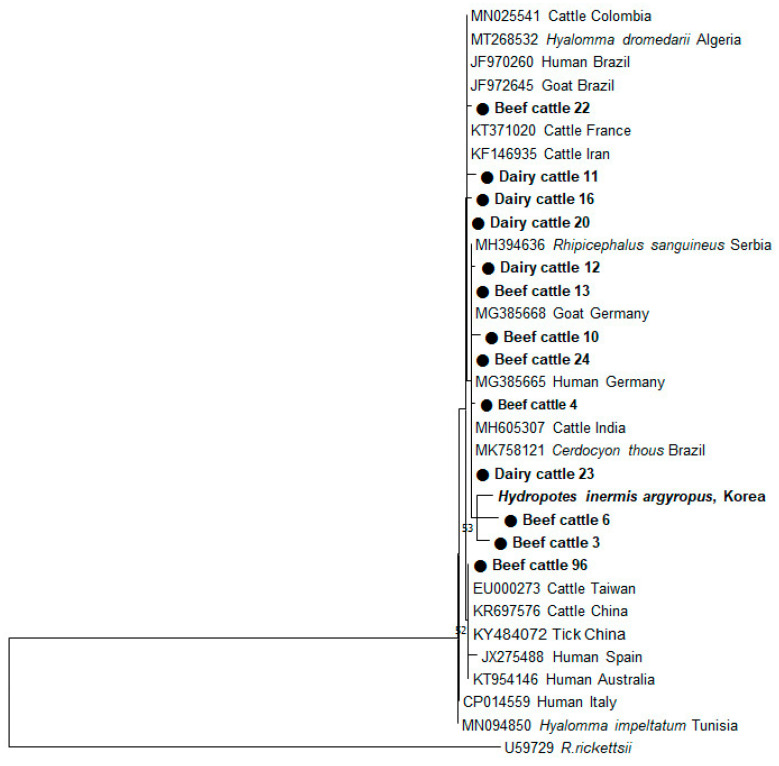
Phylogenetic analyses based on the *IS1111* sequences of *Coxiella burnetii* detected from beef and dairy cattle in the Republic of Korea (ROK). The tree was constructed using the MEGA7 software by employing the maximum-likelihood method. The numbers at the nodes of the tree indicate bootstrap values as a percentage of 1000 replicates that support each phylogenetic branch. The sequences identified in this study are marked in bold type as a circle symbol. Bootstrap values of ≥50 are shown.

**Figure 2 pathogens-09-00890-f002:**
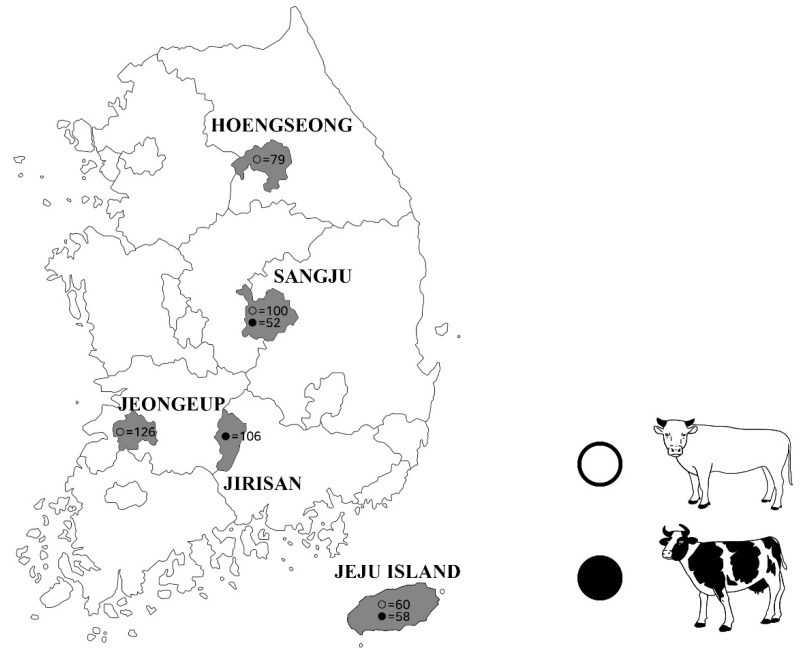
Geographical distribution of the regions from where the blood samples were collected. Beef cattle and dairy cattle are indicated using black circles and white circles, respectively, in the sampled locations.

**Table 1 pathogens-09-00890-t001:** Summary of *C. burnetii*-specific PCR and ELISA test results on growth type according to cattle breeds.

Breeds	No. of Samples	PCR (Ag)	ELISA (Ab)	PCR & ELISA
**Dairy**	**216**	**16 (7.4%)**	**3 (1.4%)**	**2 (0.9%)**
Housed	158	1 (0.6%)	1 (0.6%)	0
Grazing	58	15 (25.9%)	2 (3.4%)	2 (3.4%)
**Beef**	**275**	**37 (13.5%)**	**40 (14.5%)**	**18 (6.5%)**
Housed	136	3 (2.2%)	0	0
Grazing	139	34 (24.5%)	40 (28.8%)	18 (12.9%)
**Total**	**491**	**53 (10.8%)**	**43 (8.8%)**	**20 (4.1%)**

**Table 2 pathogens-09-00890-t002:** Prevalence of *C. burnetii* according to cattle breed and growth type.

Parameter	No. of Samples	No. of PCR Positive (%)	*P*-Value	No. of ELISA Positive (%)	*P*-Value
Beef cattle	275	37 (13.5%)	0.032	40 (14.5%)	0.000
Dairy cattle	216	16 (7.4%)		3 (1.4%)	
Grazing	197	49 (24.9%)	0.000	42 (21.3%)	0.000
Housed	294	4 (1.4%)		1 (0.3%)	
Total	491	53 (10.8%)		43 (8.8%)	

**Table 3 pathogens-09-00890-t003:** Univariate multinomial logistic regression analysis for the PCR and ELISA detection of *C. burnetii* depending on growth type.

Growth Type	PCR Positive			ELISA Positive		
	OR	*P*-Value	95% CI	OR	*P*-Value	95% CI
Housed (ref.)	1.00	–	–	1.00	–	–
Grazing	33.34	0.000	11.77–94.80	114.51	0.000	15.57–842.41

**Table 4 pathogens-09-00890-t004:** Risk factors associated with the prevalence of *C. burnetii.*

Variables		No. of *C. burnetii* Positive *	χ^2^ (*P*-Value)	OR	95% CI
Breed	Dairy cattle	17/216	−	−	−
	Beef cattle	59/275	5.82 (0.000)	3.20	1.80−5.67
Growth type	Housed	5/294	−	−	−
	Grazing	71/197	106.324 (0.000)	32.57	12.84−82.61

* PCR or ELISA positive.

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
