# Peer review of "Seroprevalence and Molecular Characterization of Coxiella burnetii in Cattle in the Republic of Korea"

_pathogens, 2020, doi:10.3390/pathogens9110890_

Round 1
Reviewer 1 Report
The authors looked for Coxiella bunertii antigens and antibodies in blood samples derived from 491 cattle and determined the presence of 53 Ag positive (by PCR) and 43 Ab positive (by ELISA test) samples. Then, they sequenced positive PCR samples and compared retrieved sequences to other previously published.
The work described follows the approach in the cited [14], but with a minor number of cattle, a less specified region of action and an additional specific focus on the growth type (as in [19]) in non-government farms not analyzed before. The prevalence of Coxiella in cattle subgroups was determined by PCR antigen detection and ELISA antibody detection. Results were expressed as positive percentages and the CI at 95% was determined assuming a normal distribution in the sample subsets and using a standard coefficient of 1.96. A P-value is computed for casual distribution, by means of chi square test. Positive PCR samples were sequenced, and 13 different sequences were used for deriving a phylogenetic tree in which they were compared against other already published Coxiella sequences. The tree is commented in lines 97-105 but fig 1 is completely lacking.
Material should be better described: providing the number and the geographic distribution of interested herds and the indicating effective consistence of the four subgroups (dairy/grazing, dairy/housing, beef/grazing, beef/housing) could add interesting information. Results, consequently, could be referred to the four subgroups and then aggregated as convenient (see also line 155). In line 196 a table 1 is cited but that seems not corresponding to the provided tab1. Methods about statistical analysis could be enriched and improved a lot to help the reader: the method followed to compute the 95% confidence interval should be added (there is a simple formula, that could be conveniently added or properly cited). The odd ratio computing, within the multivariable logistic regression model, follows a simple formula that could be added or properly cited as well.
In lines 22 and 24, the authors should follow the same scheme. As an example, if in line 22 they give data “beef versus dairy” by PCR and then by ELISA, in line 24 they should give data “grazing versus housing” by PCR and then by ELISA. In line 26 they indicate the chi square value, but it is not in the tables: it should be reported there as well. Besides the percentages 21,5 % and 7,9% reported in line 26 do not compare in tab 3 and this doesn’t help the reader.
In line 74 the value 14,5% is reported while the corresponding value in tab 1 is 14.6%, please, check for the correct value.
In line 77 and in tab 1 negative percentages appear. As they have no meaning, they should be corrected to 0%, that is the minimum possible percentage value with a biological meaning.
I encourage the authors to provide data concerning the number of AgAb ++ samples, the number of +- and the number of -+ as well.
Data in table 2 seem to be not coherent with tab 1. I tried and compute OR and gained different values. Please check. Besides, there appears the term Ag Ab positive that does not appear before. This does not help the reader. The authors should better introduce the term and use a coherent terminology in all the tables.
In line 108 the authors state that found seroprevalence (8,8%, CI 6,3%-11,3%) is lower than what previously found [14], that was 10,5%, with a CI 8,3%-12,7%. Evidently, the difference has no significance as the CI overlap. Please, check.
In discussion, some consideration should be added concerning the opposite results referred in [19]. I suggest considering the different weather conditions and growers’ behaviors that could effectively influence the bacteria’s spread indoor and outdoor.
Coxiella burnetii should be italicized wherever it appears.
In line 74 is reported: “No significance was observed between…PCR”. This seems to be in contrast with the significance threshold established at 0.05%. Please check.
Author Response
We thank the reviewers for their very careful reading and observations regarding our manuscript. The constructive comments provided were all quite valuable and helped us to prepare a suitably improved revised manuscript. We have detailed our responses to each of the points raised by the reviewers and we have modified our manuscript accordingly. Revised portions are marked in red in the text to facilitate re-review. We greatly appreciate your consideration of our revised manuscript for publication in Pathogens.
At this point, the line which the reviewer mentioned is different from the line in our manuscript.
Reviewer 1
The authors looked for Coxiella bunertii antigens and antibodies in blood samples derived from 491 cattle and determined the presence of 53 Ag positive (by PCR) and 43 Ab positive (by ELISA test) samples. Then, they sequenced positive PCR samples and compared retrieved sequences to other previously published.
The work described follows the approach in the cited [14], but with a minor number of cattle, a less specified region of action and an additional specific focus on the growth type (as in [19]) in non-government farms not analyzed before. The prevalence of Coxiella in cattle subgroups was determined by PCR antigen detection and ELISA antibody detection. Results were expressed as positive percentages and the CI at 95% was determined assuming a normal distribution in the sample subsets and using a standard coefficient of 1.96. A P-value is computed for casual distribution, by means of chi square test. Positive PCR samples were sequenced, and 13 different sequences were used for deriving a phylogenetic tree in which they were compared against other already published Coxiella sequences. The tree is commented in lines 97-105 but fig 1 is completely lacking.
We are sorry about that. At this time, we have provided. Please see Fig. 2.
Material should be better described: providing the number and the geographic distribution of interested herds and the indicating effective consistence of the four subgroups (dairy/grazing, dairy/housing, beef/grazing, beef/housing) could add interesting information.
We have provided the map which indicates regions the samples were collected. Please see new Fig. 1. Also, according to the reviewer’s comment, herds were divided into four subgroups and this information was provided as Table 1.
Results, consequently, could be referred to the four subgroups and then aggregated as convenient (see also line 155). In line 196 a table 1 is cited but that seems not corresponding to the provided tab1.
We are sorry for the confusion. We have modified all tables.
Methods about statistical analysis could be enriched and improved a lot to help the reader: the method followed to compute the 95% confidence interval should be added (there is a simple formula, that could be conveniently added or properly cited). The odd ratio computing, within the multivariable logistic regression model, follows a simple formula that could be added or properly cited as well.
We agree with this comment. We have provided the formula and corrected OR value. Please see Table 3.
In lines 22 and 24, the authors should follow the same scheme. As an example, if in line 22 they give data “beef versus dairy” by PCR and then by ELISA, in line 24 they should give data “grazing versus housing” by PCR and then by ELISA. In line 26 they indicate the chi square value, but it is not in the tables: it should be reported there as well. Besides the percentages 21,5 % and 7,9% reported in line 26 do not compare in tab 3 and this doesn’t help the reader.
We agree with this comment. We have modified as per your suggestion. We have provided the value of the chi-square in the Table 4, not in abstract.
In line 74 the value 14,5% is reported while the corresponding value in tab 1 is 14.6%, please, check for the correct value.
We agree with this comment. 14.5% is correct, so we have changed it.
In line 77 and in tab 1 negative percentages appear. As they have no meaning, they should be corrected to 0%, that is the minimum possible percentage value with a biological meaning.
I encourage the authors to provide data concerning the number of AgAb ++ samples, the number of +- and the number of -+ as well.
We have revised as per your suggestion. Please see Table 2. In addition, the data for the number of AgAb ++ samples, the number of +-, and the number of -+ Ag were provided. Please see the new Table 1.
Data in table 2 seem to be not coherent with tab 1. I tried and compute OR and gained different values. Please check. Besides, there appears the term Ag Ab positive that does not appear before. This does not help the reader. The authors should better introduce the term and use a coherent terminology in all the tables.
We agree with this comment. We have corrected it. Please see Table 3.
In line 108 the authors state that found seroprevalence (8,8%, CI 6,3%-11,3%) is lower than what previously found [14], that was 10,5%, with a CI 8,3%-12,7%. Evidently, the difference has no significance as the CI overlap. Please, check.
We simply compared the prevalence and mentioned that our values were low. Thank you for pointing out. As you comment, we have modified it to be similar.
In discussion, some consideration should be added concerning the opposite results referred in [19]. I suggest considering the different weather conditions and growers’ behaviors that could effectively influence the bacteria’s spread indoor and outdoor.
We have provided as per your suggestion. The difference between the two groups is farm management and hygiene status.
Coxiella burnetii should be italicized wherever it appears.
We have modified as per your suggestion.
In line 74 is reported: “No significance was observed between…PCR”. This seems to be in contrast with the significance threshold established at 0.05%. Please check.
We have deleted them.
Reviewer 2 Report
Line 2: Seroprevalence and molecular characterization of Coxiella burnetii in healthy cattle of the Republic of Korea
Line 16: Omit infection
Line 17: Omit cattle breed:
Line 19: Better write in the arrangement of the techniques used
Abstract: Please try to write the scientific names in italics. Moreover, I have a feeling that specific epidemiological terms are used without proper knowledge. I suggest rewriting the abstract precisely based on your specific objectives, results and conclusions with an accurate epidemiological knowledge.
Line 32: This should not be the objective of the study as its already well established that cattle do act as reservoir host for Coxiella burnetii worldwide. Also indicated by your reference 2.
Line 39: Insert living after intracellular
Line 43: Omit products as transmission from products is considered indirect
Line 45: Omit respectively
Line 56: Too selective review of literature to declare for most of the countries
Line 58: You need to synchronize the term coxiellosis and C. burnetii infection in animals throughout the manuscript
Line 63: What does it mean and how did this influence increase the meat consumption? Please write a few sentences for explanation.
Line 197: Write little more about indoor housing? How animals were fed, was there any open area? Did the animals use to move around?
Line 70: Normally serology is succeeded by PCR results. I suggest mentioning accordingly.
Line 85-86: define Ag and Ab
Line 97-98: Rephrase the sentence with the flow of the continuity in the text.
Line 102: put them to nonitalic-isolated from ticks
Line 107-116: The review of literature is selective-should be unbiased and representative
Line 133-135: Yes, but it does not guarantee the viability of the organism. To establish this fact, isolation of the organism is required in viable form. Better if you can rephrase the sentence and mention it clearly.
Line 146-152: Too uncertain statements. Better review the literature and support your arguments with published literature.
The study is interesting and might be important to some readers. Methodology is appropriate but the presentation of the results and discussion is very poor. Moreover the manuscript is flawed by very weak English. The conclusions are too general and are not supported by enough previous published literature. Author should avoid jumping to the conclusions too quickly. Also shortcomings in the study needed to be mentioned.
Author Response
We thank the reviewers for their very careful reading and observations regarding our manuscript. The constructive comments provided were all quite valuable and helped us to prepare a suitably improved revised manuscript. We have detailed our responses to each of the points raised by the reviewers and we have modified our manuscript accordingly. Revised portions are marked in red in the text to facilitate re-review. We greatly appreciate your consideration of our revised manuscript for publication in Pathogens.
At this point, the line which the reviewer mentioned is different from the line in our manuscript.
Line 2: Seroprevalence and molecular characterization of Coxiella burnetii in healthy cattle of the Republic of Korea
We have changed as per your suggestion. Please see lines 2-3.
Line 16: Omit infection
We have deleted.
Line 17: Omit cattle breed:
We have deleted.
Line 19: Better write in the arrangement of the techniques used
We have modified as per your suggestion.
Abstract: Please try to write the scientific names in italics. Moreover, I have a feeling that specific epidemiological terms are used without proper knowledge. I suggest rewriting the abstract precisely based on your specific objectives, results and conclusions with an accurate epidemiological knowledge.
We understand. Some parts of abstract were revised.
Line 32: This should not be the objective of the study as its already well established that cattle do act as reservoir host for Coxiella burnetii worldwide. Also indicated by your reference 2.
We agree with this comment. We have modified as your comment.
Line 39: Insert living after intracellular
We have added as per your suggestion.
Line 43: Omit products as transmission from products is considered indirect
We have deleted it as per your suggestion.
Line 45: Omit respectively
We have deleted it as per your suggestion.
Line 56: Too selective review of literature to declare for most of the countries
We have deleted it as per your suggestion.
Line 58: You need to synchronize the term coxiellosis and C. burnetii infection in animals throughout the manuscript
We used the term “C. burnetii infection”, not coxiellosis.
Line 63: What does it mean and how did this influence increase the meat consumption? Please write a few sentences for explanation.
We have provided as per your suggestion. Please see the introduction.
Line 197: Write little more about indoor housing? How animals were fed, was there any open area? Did the animals use to move around?
We have provided as per your suggestion. Please see lines 220-226.
Line 70: Normally serology is succeeded by PCR results. I suggest mentioning accordingly.
In order to avoid ambiguity, we have deleted this sentence.
Line 85-86: define Ag and Ab
Sorry to be confused. Ag is PCR detection and Ab is ELISA serology. We have replaced Ag and Ag with PCR and ELISA.
Line 97-98: Rephrase the sentence with the flow of the continuity in the text.
We have modified as per your suggestion. Please see lines 106-108.
Line 102: put them to nonitalic-isolated from ticks
We have modified it.
Line 107-116: The review of literature is selective-should be unbiased and representative
We have provided in more detail. Please see lines 120-126.
Line 133-135: Yes, but it does not guarantee the viability of the organism. To establish this fact, isolation of the organism is required in viable form. Better if you can rephrase the sentence and mention it clearly.
In order to avoid ambiguity, we have modified this sentence and provided some. Please see lines 141, and 147-149.
Line 146-152: Too uncertain statements. Better review the literature and support your arguments with published literature.
We have provided some cites which support our thoughts. Please see lines 160, 161,163, and 168-172.
Reviewer 3 Report
This is a well-written manuscript describing a study of C. burnetii infection and seroprevalence in the Republic of Korea. The work is pithy and results are clearly described and demonstrated in text and tables.
Line 38 – Organisms name should be italicized throughout the manuscript.
Line 41 – Goat and sheep are more through to be the primary source of human infection, with cattle less so.
Line 43 – add “,” after dust and delete “in nature”
Line 46 – discharge should be singular
Line 54 – I am not certain I understand this sentence. “in fields” does not make sense. Please rewrite this sentence
Line 70-71 – remove “considered”, replace “for C. burnetii antibodies” with “ELISA serology”
Line 77 and table, etc. – replace “housing” with “housed”
Line 102 – remove italics from “isolated from ticks”
Line 110 – if possible, list the countries and prevalences, please as this is informative
Line 111-112 – is this also reference 14, is so please cite
Line 117-127 – Was there any serology or PCR performed on calves? It may be interesting if this is possible to test from cached samples, and if not, potentially this could be examined in a future study.
Author Response
We thank the reviewers for their very careful reading and observations regarding our manuscript. The constructive comments provided were all quite valuable and helped us to prepare a suitably improved revised manuscript. We have detailed our responses to each of the points raised by the reviewers and we have modified our manuscript accordingly. Revised portions are marked in red in the text to facilitate re-review. We greatly appreciate your consideration of our revised manuscript for publication in Pathogens.
At this point, the line which the reviewer mentioned is different from the line in our manuscript.
Reviewer 3
This is a well-written manuscript describing a study of C. burnetii infection and seroprevalence in the Republic of Korea. The work is pithy and results are clearly described and demonstrated in text and tables.
Line 38 – Organisms name should be italicized throughout the manuscript.
We thoroughly revised and changed as per your comment.
Line 41 – Goat and sheep are more through to be the primary source of human infection, with cattle less so.
We agree with this comment. We have deleted these.
Line 43 – add “,” after dust and delete “in nature”
We have revised as per your comment. Please see line 55.
Line 46 – discharge should be singular
We have revised as per your comment. Please see line 58.
Line 54 – I am not certain I understand this sentence. “in fields” does not make sense. Please rewrite this sentence
We agree with this comment. We have clearly mentioned. Please see line 67.
Line 70-71 – remove “considered”, replace “for C. burnetii antibodies” with “ELISA serology”
We agree with this comment. We have revised as per your comment. Please see line 90.
Line 77 and table, etc. – replace “housing” with “housed”
We have modified the whole manuscript including all tables as per your suggestion.
Line 102 – remove italics from “isolated from ticks”
We have modified as per your comment.
Line 110 – if possible, list the countries and prevalences, please as this is informative
We agree with this comment. We have provided as per your comment. Please see lines 120-122.
Line 111-112 – is this also reference 14, is so please cite
We agree with this comment. We have provided as per your comment.
Line 117-127 – Was there any serology or PCR performed on calves? It may be interesting if this is possible to test from cached samples, and if not, potentially this could be examined in a future study.
Unfortunately, we did not test Q fever on calves. Maybe, we will investigate on future study.
Round 2
Reviewer 1 Report
The authors, after the suggestions of the three revisors, significatively improved the paper. Nonetheless, signals suggesting hurry or lack of care are still present. Here I list some flaws still present in the paper that must be corrected prior to publication.
In general, a careful checking of spelling and grammar is needs as some refuses are still present.
Fig 1 has been added as requested, but the bootstrap values (cited in figure caption and in the methods) are still lacking in the figure, so that nothing can be told about some subgroups present in the tree.
Methods about statistical analysis have been enriched and improved but there is still confusion between univariate and multivariate (please, check the correct words) logistic regression. In my understanding, no multivariate logistic regression has been afforded as no combined odd ratios (combined risk increasing by grazing of beef cattle) are provided. Please check for a better understanding in methods.
Last line in pag 2 can be still improved: as p-value expresses more precisely the same data of CI95%, I would suggest using the same scheme used in the precedent line not including CI 95% values and including the p-value.
In discussion, some consideration have been added concerning the opposite results referred in [19] but comments are still lacking about light discrepancies found between ELISA and PCR (tab3) and their combined use (Tab 4, where all samples ++, +- and -+ are equally considered as positive).
Formatting is still incorrect.
Author Response
We thank the reviewers for their very careful reading and observations regarding our manuscript. The constructive comments provided were all quite valuable and helped us to prepare a suitably improved revised manuscript. We have detailed our responses to each of the points raised by the reviewers and we have modified our manuscript accordingly. Revised portions are marked in red in the text to facilitate re-review. We greatly appreciate your consideration of our revised manuscript for publication in Pathogens.
The authors, after the suggestions of the three revisors, significatively improved the paper. Nonetheless, signals suggesting hurry or lack of care are still present. Here I list some flaws still present in the paper that must be corrected prior to publication.
In general, a careful checking of spelling and grammar is needs as some refuses are still present.
We read the manuscript thoroughly and revised spelling and grammar. We think that this manuscript has been improved.
Fig 1 has been added as requested, but the bootstrap values (cited in figure caption and in the methods) are still lacking in the figure, so that nothing can be told about some subgroups present in the tree.
We agree with this comment. We have provided the bootstrap values. Please see the line 115 and a modified a figure 1.
Methods about statistical analysis have been enriched and improved but there is still confusion between univariate and multivariate (please, check the correct words) logistic regression. In my understanding, no multivariate logistic regression has been afforded as no combined odd ratios (combined risk increasing by grazing of beef cattle) are provided. Please check for a better understanding in methods.
We agree with this comment. We made a mistake and corrected as your suggestion. Please see lines 270-272.
Last line in pag 2 can be still improved: as p-value expresses more precisely the same data of CI95%, I would suggest using the same scheme used in the precedent line not including CI 95% values and including the p-value.
We have modified as your suggestion. Please see Table 2.
In discussion, some consideration have been added concerning the opposite results referred in [19] but comments are still lacking about light discrepancies found between ELISA and PCR (tab3) and their combined use (Tab 4, where all samples ++, +- and -+ are equally considered as positive).
We made a mistake and corrected the sentences. Please see lines 150-158.
Formatting is still incorrect.
At this time, we fit the manuscript into a journal format.
Reviewer 2 Report
The authors have tried to improve substantially but still there are lots of general and nontechnical statements especially in discussion. The review of literature in first paragraph of discussion is very selective. I can see reports from other countries in my database. Moreover, you mentioned isolates of Coxiella burnetii; did you isolate them, if not-as I see it from your manuscript, this term should be avoided as PCR products can not be regarded as isolates at any case. Further more you have used italics and non italics in text without any valid reason e.g. in the whole manuscript you have mentioned Coxiella burnetii without italics many times. Should be streamlined. Conclusions are not genuine as based on PCR you can not declare animal as a reservoir host-you need to isolate from those animals.
Author Response
We thank the reviewers for their very careful reading and observations regarding our manuscript. The constructive comments provided were all quite valuable and helped us to prepare a suitably improved revised manuscript. We have detailed our responses to each of the points raised by the reviewers and we have modified our manuscript accordingly. Revised portions are marked in red in the text to facilitate re-review. We greatly appreciate your consideration of our revised manuscript for publication in Pathogens.
The authors have tried to improve substantially but still there are lots of general and nontechnical statements especially in discussion. The review of literature in first paragraph of discussion is very selective. I can see reports from other countries in my database. Moreover, you mentioned isolates of Coxiella burnetii; did you isolate them, if not-as I see it from your manuscript, this term should be avoided as PCR products can not be regarded as isolates at any case. Further more you have used italics and non italics in text without any valid reason e.g. in the whole manuscript you have mentioned Coxiella burnetii without italics many times. Should be streamlined. Conclusions are not genuine as based on PCR you can not declare animal as a reservoir host-you need to isolate from those animals.
We agree with this comment. As your suggestion, we have provided the prevalence by countries which we can find, replaced isolates with sequences, used italics, and then deleted the sentence which you mentioned. In addition, we fit the manuscript into a journal format and it’s easy to read. Please see lines 101-107, 114, 120-122, and 196-197.
Round 3
Reviewer 2 Report
The authors have substantially revised and tried to improve the scientific content of the paper but still the author seems confused between specific epidemiological terms e.g. sero-prevalence, sero-positivity, infection and prevalence. Author has mixed these terms repeatedly throughout the manuscript and does not seem to know the valid interpretation of the tests used. Moreover, English language is not to a satisfactory level. Although the topic is very interesting and results/findings may be important for a significant number of readers rephrasing of the results and discussion seems necessary along with English language improvements.
Author Response
We thank the reviewers for their very careful reading and observations regarding our manuscript. The constructive comments provided were all quite valuable and helped us to prepare a suitably improved revised manuscript. We have detailed our responses to each of the points raised by the reviewers and we have modified our manuscript accordingly. Revised portions are marked in red in the text to facilitate re-review. We greatly appreciate your consideration of our revised manuscript for publication in Pathogens.
The authors have substantially revised and tried to improve the scientific content of the paper but still the author seems confused between specific epidemiological terms e.g. sero-prevalence, sero-positivity, infection and prevalence. Author has mixed these terms repeatedly throughout the manuscript and does not seem to know the valid interpretation of the tests used. Moreover, English language is not to a satisfactory level. Although the topic is very interesting and results/findings may be important for a significant number of readers rephrasing of the results and discussion seems necessary along with English language improvements.
We thoroughly revised the manuscript according to the reviewer’s comment and performed English proofreading by an expert. We think that the manuscript has greatly improved.